# Effect of Acetylated SEBS/PP for Potential HVDC Cable Insulation

**DOI:** 10.3390/ma14071596

**Published:** 2021-03-25

**Authors:** Peng Zhang, Yongqi Zhang, Xuan Wang, Jiaming Yang, Wenbin Han

**Affiliations:** 1Key Laboratory of Engineering Dielectrics and Its Application, Ministry of Education, Harbin University of Science and Technology, Harbin 150080, China; zp199620@163.com (P.Z.); zhangyongqihust@163.com (Y.Z.); 15344514800@163.com (W.H.); 2School of Electrical and Electronic Engineering, Harbin University of Science and Technology, Harbin 150080, China

**Keywords:** polypropylene, voltage stabilizer, HVDC cable, thermoplastic elastomer

## Abstract

Blending thermoplastic elastomers into polypropylene (PP) can make it have great potential for high-voltage direct current (HVDC) cable insulation by improving its toughness. However, when a large amount of thermoplastic elastomer is blended, the electrical strength of PP will be decreased consequently, which cannot meet the electrical requirements of HVDC cables. To solve this problem, in this paper, the inherent structure of thermoplastic elastomer SEBS was used to construct acetophenone structural units on its benzene ring through Friedel–Crafts acylation, making it a voltage stabilizer that can enhance the electrical strength of the polymer. The DC electrical insulation properties and mechanical properties of acetylated SEBS (Ac-SEBS)/PP were investigated in this paper. The results showed that by doping 30% Ac-SEBS into PP, the acetophenone structural unit on Ac-SEBS remarkably increased the DC breakdown field strength of SEBS/PP by absorbing high-energy electrons. When the degree of acetylation reached 4.6%, the DC breakdown field strength of Ac-SEBS/ PP increased by 22.4% and was a little higher than that of PP. Ac-SEBS, with high electron affinity, is also able to reduce carrier mobility through electron capture, resulting in lower conductivity currents in SEBS/PP and suppressing space charge accumulation to a certain extent, which enhances the insulation properties. Besides, the highly flexible Ac-SEBS can maintain the toughening effect of SEBS, resulting in a remarkable increase in the tensile strength and elongation at the break of PP. Therefore, Ac-SEBS/PP blends possess excellent insulation properties and mechanical properties simultaneously, which are promising as insulation materials for HVDC cables.

## 1. Introduction

High-voltage direct current (HVDC) cable systems are used globally to transmit high power for long distances when the connection to high-voltage alternating current cable systems is not feasible or economically convenient [1]. DC power transmission has the advantages of high capacity, excellent stability, and low power consumption [2,3,4,5]. Cross-linked polyethylene (XLPE) is widely used in HVDC cable insulation because of its excellent dielectric and thermomechanical properties. However, as a thermo-set material, XLPE is challenging to recycle at the end of its useful life, and incinerating the waste not only causes environmental pollution but also wastes resources. Nowadays, it is more important than ever to reduce the impact of human activities on the environment. Therefore, the search for environmentally friendly insulation materials with excellent performance has currently become mainstream [6].

Polypropylene (PP) is a thermoplastic material with wide sources, low prices, and easy recycling, which not only meets both economic and environmental needs but also has a high melting point and excellent electrical and mechanical properties. As an environmentally friendly cable insulation material, PP can replace the current XLPE material to some extent [7,8,9]. However, PP cannot meet the requirements of mechanical toughness of insulation materials for HVDC cables because of its large elastic modulus and poor impact resistance. Adding elastomer to PP is a simple and effective toughening method. However, it has been found that the blending of PP and elastomer not only decreases the electrical strength of PP but also intensifies the space charge injection, making it unable to meet the requirements of electrical strength for HVDC cables [10,11]. Therefore, how to achieve excellent electrical properties and mechanical toughness of PP at the same time has become an urgent problem to be solved.

Currently, nano additive filling or organic functional group grafting methods are widely used to enhance the electrical properties of polymers [12,13,14,15]. Zha et al. [16] found that doping nano-ZnO into a PP/SEBS composite system not only improved the electrical resistance of the composite system but also inhibited space charge injection. In a study of SiO_2_/polyolefin elastomer (POE)/PP composite systems, Chi et al. [17] found that SiO_2_ also enhanced the electrical properties of the composite systems. Zha et al. [18] found that the space charge suppression ability of modified PP was significantly enhanced by PP grafting with maleic anhydride (MAH) and consequently improved the breakdown field strength of PP. However, in actual production, to ensure the dispersion of nanoparticles and the grafting efficiency of organic functional groups, both need to mix the composites thoroughly for a long time. However, the presence of many tertiary carbon atoms in the molecular structure of PP and elastomers tends to degrade PP and elastomers under the effect of continuous heat and shear forces, which eventually cannot be applied to HVDC cable insulation. It has been reported that the electrical strength of polymers can be effectively improved by filling with voltage stabilizers [19,20,21]. Acetophenone and its derivatives are the earliest reported voltage stabilizers. Under strong electric fields, acetophenones will be first impacted by electrons and excited or ionized due to their high electron affinity energy, and the secondary electrons obtained are of low energy and less likely to redamage the molecular chains of polymers, thus improving the electrical resistance of insulating materials [22]. Li et al. [23] found that the addition of 0.4 phr acetophenone could increase the DC breakdown strength of PE by 30%. However, small molecule acetophenone is poorly compatible with polymer, and acetophenone tends to migrate out of the matrix during the use of the cable and cause environmental pollution. Studies have shown that acetophenone also causes a significant increase in the anisotropic space charge in the insulation and increases the electrical conductivity of the insulation material [24]. Therefore, acetophenone cannot be directly applied to polypropylene/elastomer composite systems. Currently, there is no feasible method to improve the electrical strength of PP/elastomer.

If the elastomer can be combined with acetophenone, it can not only avoid the migration of acetophenone small molecules from the polymer system but also improve the insulating property and mechanical toughness of the polymer. Zhang et al. [25,26] found through theoretical calculations that acetophenone and its derivatives can undergo a keto-enol exchange isomerization reaction after absorbing energy, and acetophenone grafting to polyethylene chains can also improve the electrical strength of polyethylene. Dong et al. [27] synthesized a grafted voltage stabilizer that can be grafted onto XLPE molecule chains during the crosslinking process. It can improve the breakdown strength and suppress the migration of voltage stabilizers. Yamano [28] found that deep traps can also be introduced when polar groups are found on benzene ring compounds, which have the effect of inhibiting space charge.

SEBS is a new type of thermoplastic elastomer that can be used at high temperatures. In addition, SEBS can be recycled, which is in line with the concept of environmental protection [29,30,31]. Based on this, in this study, SEBS was acetylated to produce acetophenone structural units through Friedel–Crafts acylation, and then the acetylated SEBS (Ac-SEBS) was melt blended with PP to investigate the electrical insulation properties and mechanical properties of the blends and to determine whether the blends could be used in HVDC cables. It was shown that the acetophenone unit modified in SEBS has the effect of the voltage stabilizer and can remarkably enhance the DC breakdown strength of the SEBS/PP. Meanwhile, Ac-SEBS carries polar groups, which can introduce deep traps in the material system, and to a certain extent can inhibit the accumulation of space charges in the blends and decrease the conductivity of the material by trapping the charge. Besides, Ac-SEBS still has the toughening effect of SEBS, which can enhance the elongation at break and tensile strength of PP. The results show that Ac-SEBS/PP possesses both an excellent electrical and mechanical performance and is expected to be applied to HVDC cable insulation.

## 2. Materials and Methods

### 2.1. Materials

PP (T30S) was purchased from Sinopec (Beijing, China). SEBS (g1652) was purchased from KRATON (Houston, TX, USA). Dichloromethane, methanol, and n-heptane were purchased from Tianjin Fuyu Fine Chemical Co, Ltd. (Tianjin, China). Acetyl chloride and anhydrous aluminum chloride were purchased from Shanghai Aladdin Biochemical Technology Co., Ltd. (Shanghai, China). The antioxidant 1010 was produced by Dongguan Yamaichi Plastic Chemical Co., Ltd. (Dongguan, China).

Ac-SEBS was prepared according to the Friedel–Crafts acylation, and the chemical equation for the synthesis of Ac-SEBS is shown in Figure 1. In total, 10 g of SEBS was dissolved in 200 mL of dichloromethane, and then the mixture was heated to 75 °C to dissolve all the SEBS. When the solution was cooled to room temperature, 5 mL of acetyl chloride were added to the solution, and after stirring for 5 min, a certain mass (0.5, 0.7 g) of anhydrous aluminum chloride was added to make the reaction proceed for 3 h. After the reaction, 300 mL of methanol were poured into the solution to precipitate white flocculent precipitates. The precipitates were filtered and dried in an oven at 50 °C. Then, the precipitates were dissolved in 200 mL of dichloromethane, heated to 75 °C, and dissolved completely. Then, 300 mL of methanol were slowly poured into the solution to precipitate white flocculent precipitates. The precipitates were filtered and dried in an oven at 50 °C to obtain pure Ac-SEBS. 

Ac-SEBS, PP, and antioxidants were added to the torques rheometer for melting blending, in which the temperature was set at 190 °C, the rotor speed was set at 60 rpm, and the mixing was 5 min, in which the content of Ac-SEBS was 30%, and the content of antioxidants was 1%. After evenly mixed samples were taken out, according to different experimental requirements, a flat-plate vulcanizing machine was applied to press the samples into sheets of different thicknesses at 190 °C and 15 MPa for 15 min.

### 2.2. Characterization and Testing Scheme

The Fourier-transform infrared (FT-IR) spectroscopy of SEBS and Ac-SEBS was tested to detect the results of acetylation on a Nicolet iS5 spectrometer (Madison, WI, USA). The tested wavenumber range was 400–4000 cm^−1^ with a resolution of 4 cm^−1^ and a total of 32 scans. The nuclear magnetic resonance hydrogen (^1^H NMR) spectra were measured at 400 MHz (^1^H) on a Bruker AVANCE III (Karlsruhe, Germany) to determine the degree of acetylation of Ac-SEBS obtained after the addition of different masses of anhydrous aluminum chloride. CDCl_3_ was used as the solvent, and tetramethylsilane (TMS) was used as the internal standard. The microscopic morphology of Ac-SEBS/PP at a low accelerating voltage of 5 kV was observed on scanning electron microscopy (SEM, SU8020, Tokyo, Japan). The specimens were fractured in liquid nitrogen and then etched with n-heptane at room temperature. After washing and drying, the specimens were attached to a sample stage, and the surface was sprayed with gold to observe the morphology of the specimens. 

To test the thermal stability of the samples, the TG209 F3 thermogravimetric analyzer (Bavaria, Germany) was used to perform the thermogravimetric analysis (TGA). The specimens were placed in a platinum crucible and heated from 50 to 600 °C at a rate of 10 °C /min under the protection of a nitrogen atmosphere.

The tensile test was performed on a functional electronic tensile tester (CMT6000) produced by Meitesi Industry System Co., Ltd. (Shanghai, China). A dumbbell-shaped specimen with a thickness of 1 mm, a width of 4 mm, and a gauge length of 20 mm was used. The test was conducted at 25 °C with a stretching rate of 50 mm/min according to ASTMD 638-2003. Each group of samples was tested 5 times, and the average value was calculated. DC breakdown experiments were performed using a DC breakdown system (Suzhou, China) with a uniform ramp-up rate of 1 kV/s until the material with a thickness of 50 µm was broken down. The specimens and the electrode were submerged in silicone oil during the experiments to avoid flashover along the edge. The voltage value at the time of breakdown was read, and the corresponding DC breakdown strength was calculated according to the formula E = U/d, where U is the voltage of the specimens at the time of breakdown, d is the thickness of the specimens, and E is the breakdown field strength of the specimens. The two-parameter Weibull statistical distribution method was used to process the results of 12 tests for each specimen, and the breakdown field strength with a cumulative damage probability of 63.2% was taken as the characteristic breakdown field strength of the material. 

The space charge distribution of the test specimens at room temperature was measured by the pulsed electro-acoustic (PEA) system (Shanghai, China). The test method is to polarize the sample under an electric field strength of 40 kV/mm for 30 min to obtain the spatial charge distribution in the process of polarization. Then, the sample was short-circuited to measure the distribution and change of space charge in the sample during the process of 30-min depolarization. The sample to be measured was a circular specimen with a diameter of 60 mm and a thickness of 265 ± 20 μm. The DC conductive current of the sample was measured by a three-electrode measurement system, in which the lower electrode was connected to DC high voltage, the protective electrode was grounded, and the measuring electrode was connected to the electrostatic meter through a protective resistor with a resistance of 50 MΩ. The thickness of the sample was 0.2 mm, and an aluminum electrode with a vacuum evaporation radius of 50 mm was used as the measuring electrode on one side, and the protective gap was about 2 mm. The electric field strength was increased from 4 to 40 kV/mm, and the conductive current was recorded for 20 min each time, and then the field strength was increased. Finally, the conductive current density under different electric field strengths was calculated according to the formula J = I/S, where J represents the current density, I represents the current, and S represents the specimen area.

## 3. Results and Discussion

### 3.1. Structural Characterization

Figure 2 shows the IR spectra of SEBS and Ac-SEBS. Compared with SEBS, Ac-SEBS showed new absorption peaks at 1684, 1269, and 826 cm^−1^. The absorption peak at 1684 cm^−1^ represents the C=O stretching vibration, the absorption peak at 1269 cm^−1^ represents the vibration of the aromatic ketone skeleton, and the weak absorption peak at 826 cm^−1^ represents the para substitution of the benzene ring for the C-H surface bending vibration [32]. Therefore, the infrared spectrum can prove that the acetyl group successfully grafted to the para position of the benzene ring in the polystyrene block of SEBS after the acetylation reaction and the acetophenone structural unit is present on Ac-SEBS. 

Since the area of the resonant peak in the ^1^H NMR spectra is proportional to the number of protons generated in the peak, the acetylation degree can be calculated according to the ratio of the proton peak area. From Figure 3, it was found that the benzene ring proton resonance of SEBS and Ac-SEBS occurred mainly in three regions (A, 6.3–6.8 ppm, B, 6.8–7.2 ppm, and C, 7.4–7.7 ppm). Region A represents the ortho proton peak of the benzene ring in the polystyrene block on SEBS. Region B represents the meta and para proton absorption peaks of benzene rings in SEBS. In the polystyrene blocks, some of the neighboring hydrogen nuclei on the benzene ring split under the influence of the electron shielding effect on the surrounding benzene ring, so the chemical shifts of region A and region B are different. Region C represents the absorption peak formed when the benzene ring is acetylated, and the carbonyl group is π-π conjugated. The two proton absorption peaks adjacent to the carbonyl group on the benzene ring migrate to the low field. Region D (2.4–2.6 ppm) represents the proton peak of the methyl group of the acetyl group on Ac-SEBS. The number of protons represented by region A does not change with the acetylation of SEBS, while the number of protons represented by region D increases with the increase of the degree of acetylation. Therefore, the acetylation degree of SEBS can be calculated by the ratio of the area of peak A to the area of peak D [32]. The acetylation degree of the samples could be calculated by Equation (1):(1)$$\mathrm{Sub}\%=\frac{2\times A_{D}}{3\times A_{A}}\times 100\%\;$$
where Sub is the acetylation degree of Ac-SEBS, A_A_ is the area of the proton peak of region A, and A_D_ is the area of the proton peak of region D. A_D_/3 represents the number of methyl groups in the acetophenone group, and A_A_/2 represents the number of benzene rings in the SEBS.

The microstructure of the samples was observed by SEM. Figure 4 shows the cross-sectional structure of PP and its blends. The holes in the figure represent the SEBS etched by n-heptane, so the distribution of the holes can represent the dispersion of SEBS and Ac-SEBS in PP. The SEBS contains block copolymerization of polystyrene units, which disrupts the regularity of the macromolecular structure, and the material is in an amorphous aggregated state, so PP and elastomer cannot be homogeneously blended. Compared to the PP matrix, the blends exhibit a distinct sea-island structure. Among them, PP is the continuous phase, and the elastomer is the dispersed phase. From the figure, it can be seen that when the acetylation degree of Ac-SEBS is low, the pore size is small and the number is small, and Ac-SEBS can be uniformly distributed in polypropylene with almost no agglomeration, which indicates that PP is compatible with Ac-SEBS to some extent but not completely. However, the number of holes gradually increases as the degree of acetylation increases, and the agglomeration phenomenon is very obvious when the degree of acetylation reached 12%. This is because as the acetylation degree increases, the molecular weight of Ac-SEBS increases, the number of polar groups increases, and both the interaction of polar groups and the increasing molecular weight of Ac-SEBS make the blends less compatible, which eventually leads to a large amount of agglomeration of Ac-SEBS in the polypropylene matrix.

### 3.2. Thermal Stability

Figure 5 shows the thermogravimetric curves of PP, SEBS/PP, and Ac-SEBS/PP. It can be seen from Figure 5 that the thermal stability of Ac-SEBS /PP and SEBS/PP is better than PP, and the extrapolated onset temperatures of PP, SEBS/PP, 4.6% AC-SEBS, and 12% AC-SEBS are 446.0, 456.0, 457.1, and 457.0 °C, respectively. The reason that the thermal stability of AC-SEBS /PP and SEBS/PP is better than PP is that the introduction of SEBS and AC-SEBS inhibits the molecular chain movement of PP, thus increasing the intermolecular force. This results in an increase in the energy required to break the main chains of macromolecules during heating, leading to an improvement in heat resistance. Moreover, the thermal stability of AC-SEBS /PP is similar to that of SEBS/PP, indicating that the construction of acetophenone structural units on SEBS could effectively inhibit the migration of acetophenone.

### 3.3. Stress–Strain Curve

Figure 6 shows the stress–strain curves of PP, SEBS/PP, and Ac-SEBS/PP. The tensile strength and elongation at break of different specimens are shown in Table 1. From Figure 6, it is found that the elongation at break and tensile strength of PP is low, and after the incorporation of 30% SEBS, the elongation at break and tensile strength increase substantially, and the yield point also shifts to the high strain. Compared with SEBS/PP, the elongation at break and tensile strength of Ac-SEBS/PP decrease, and with the increase of acetylation, the tensile strength of the samples gradually increases, and the elongation at break gradually decreases. 

It was found that PP possesses excellent elastic modulus yet also has poor toughness [33]. Therefore, the elongation at break and the tensile strength of PP are both low. SEBS has a flexible ethylene-butene block with high elasticity of rubber at room temperature, which can toughen PP, so the elongation at break of PP increase and the increase in tensile strength may be due to the entanglement between PP molecular chains and SEBS flexible molecular chains, which form more physical entanglement points and hinder the fracture of the specimens. However, more polar groups are introduced on Ac-SEBS, which inhibits the movement of molecular chains. As the degree of acetylation increases, more polar groups are introduced, and the inhibitory effect on the movement of molecular chains is more obvious so that the flexibility of SEBS gradually decreases, leading to a gradual decrease in the elongation at break. When the degree of acetylation is low, a small number of acetyl groups are introduced into the side chains of SEBS, which increases the intermolecular distance and leads to a decrease the of intermolecular force, resulting in the decrease of the tensile strength of the specimens. As the acetylation degree increases, the molecular weight of the specimens increases greatly, and the flexibility of the SEBS molecular chain becomes worse, which makes the intermolecular force increase and finally makes the tensile strength gradually increase. The experimental results show that Ac-SEBS can still enhance the toughness of PP and can make PP meet the mechanical property requirements of HVDC cable.

### 3.4. DC Breakdown Strength

Figure 7 shows the DC breakdown strengths of PP, SEBS/PP, and Ac-SEBS/PP. The shape parameter can correspond to the dispersion of the data, and the scale parameter can correspond to the DC breakdown field strength of the specimens. As can be seen in Figure 7, the breakdown strength of PP decreased from 310.8 to 256.6 kV/mm, with a decrease of 17.4% after adding 30% SEBS to PP. The reason for the decrease in breakdown strength may be that PP and SEBS are not fully compatible and the incorporation of a large number of SEBS introduces a large number of interfaces, which causes severe interface loss and partial discharge. In addition, the introduction of SEBS increases the inhomogeneity of the blends and increases the free volume of the blends so that the free path of electrons increases under high electric fields, which promotes the accumulation of high-energy electron energy and leads to a significant decrease in the breakdown field strength of PP. 

However, the DC breakdown strength of PP/Ac-SEBS is higher than that of PP/SEBS, and as the acetylation degree increases, the influence of Ac-SEBS on the breakdown strength of PP decreases. The increase of the DC breakdown strength is because Ac-SEBS plays the role of a voltage stabilizer, and after constructing an acetophenone group with high electron affinity on SEBS, under the strong electric field, Ac-SEBS will first be impacted by electrons, and at the same time, the keto-enol isomer exchange reaction of the acetophenone structural unit absorbs high-energy electrons to consume energy and releases it with relatively harmless energy, which reduces the possibility of damage to the PP molecular chain by high-energy electrons. In addition, Ac-SEBS has a polar group carbonyl group, which can enhance the electron scattering effect and thus improve the DC breakdown field strength of the blend system. However, when the acetylation degree is too high, the interaction between the polar groups makes the blend system less compatible, and the structure of the blend system is more inhomogeneous, which aggravates the partial discharge and causes severe electric field distortion. When the degree of acetylation reached 12%, the DC breakdown strength of the blend system is even lower than that of SEBS/PP. At this time, the interaction between Ac-SEBS and PP is significantly weakened, and Ac-SEBS lose its role as a voltage stabilizer and is equivalent to an impurity in the PP matrix, leading to the final decrease of the breakdown strength. The results show that Ac-SEBS with the appropriate degree of acetylation can play the role of a voltage stabilizer and enhance the breakdown strength of the blends.

### 3.5. Space Charge Characteristics

Figure 8 shows the space charge distribution of the sample during polarization and short-circuit. As seen from Figure 8, under the electric field strength of 40 kV/mm, only a small amount of space charge accumulates in the PP. After the introduction of SEBS, a large number of hetero space charges are generated near the cathode and anode, which is due to the fact that PP is a nonpolar polymer with a relatively regular structure, which makes charge injection more difficult and space charge cannot accumulate in large amounts, while the introduction of SEBS makes the structure inhomogeneous. Under the electric field, the positive and negative polar charges formed by electrode injection and dissociation of small molecule impurities in the specimen migrate to opposite directions, and the charge transfer in the specimen is blocked by the interfacial barrier formed by PP/SEBS, resulting in charge accumulation at the interface. 

Compared with SEBS/PP, the space charge in 4.6%Ac-SEBS/PP is obviously suppressed, and according to the distribution of space charge when short-circuited between SEBS/PP and 4.6%Ac-SEBS/PP, the maximum accumulation of space charge in SEBS/PP is 5.38 C/m^3^ and decays to 4.08 C/m^3^ after 30 min, while the space charge in 4.6%Ac- SEBS/PP decays from 2.78 to 2.09 C/m^3^, which is significantly less than that of SEBS/PP. This is probably because Ac-SEBS contains the polar groups, carbonyl, and the polar groups introduce more deep traps in PP, which are difficult, for electrons, to escape from after being trapped. These traps will have a strong restraint effect on the injected charge and restrict its migration to the interior of the material, thus reducing the amount of space charge inside the material. Moreover, AC-SEBS has a high electron affinity, and the deep trap inside can more easily capture the charge injected from the electrode, and the charge that is difficult to escape forms an independent electric field near the electrode and changes its effective electric field. At this time, the trapped charge exhibits a retarding effect on charge injection, and increases the potential barrier of the charge injected from the electrode, and ultimately reduces the carrier mobility [15]. When the degree of acetylation reaches 12%, the space charge cannot be suppressed, and the charge decay rate is accelerated during short-circuiting. This is because the agglomeration of Ac-SEBS in the PP matrix increases the inhomogeneity of the blend system, the interaction between Ac-SEBS and the polymer matrix is weakened, and some of the deep traps originally introduced by Ac-SEBS become shallow traps. Additionally, some of the charges easily escape from the traps, which instead makes it easier for the charges to migrate to the interior of the specimen, resulting in severe space charge accumulation. In conclusion, Ac-SEBS with the appropriate degree of acetylation has the effect of suppressing space charge.

### 3.6. Electrical Conductivity

Figure 9 shows the relationship between the conductive current density of PP and its blends with the change of electric field strength. As seen in Figure 9, the curve can be divided into two parts of each sample. Under low electric field strength, the charge conduction in the specimens is dominated by ohmic conductivity, and when the electric field strength reaches a specific value, the concentration of the injected carriers increases, and a severe accumulation of space charge occurs, causing a space charg-limited current (SCLC), which causes the current through the dielectric to shift from the ohmic current region to the SCLC region, at which time the slope of the curve increases. 

Under lower electric field strengths, the conductive current density of SEBS/PP is smaller than that of PP, which is due to the fact that the interface composed of SEBS and PP can trap a large amount of charge under the effect of factors, such as the interfacial potential barriers and surface states, which inhibit the migration of the carrier [34]. However, the introduction of SEBS leads to considerable accumulation of space charge in the blends, which makes SEBS/PP enter the SCLC region at the lower electric field strength. When the degree of acetylation reaches 4.6%, the current density of Ac-SEBS/PP is always lower than that of PP and SEBS/PP, probably due to the strong electron affinity of the acetophenone structural unit in Ac-SEBS, which has the effect of capturing electron. Its molecules cannot move under the electric field, thus inhibiting carrier directional migration and reducing the electronic conductivity, which in turn decreases the overall conductivity of the material. Moreover, because of its inhibitory effect on space charge, the inflection point of the ohmic current region to the SCLC region shifts to the high electric field. When the degree of acetylation is too high, Ac-SEBS becomes less compatible with PP, and the interaction between Ac-SEBS and the PP matrix is weakened, at which time, the capture effect of Ac-SEBS on electrons can hardly affect the conductive current density of PP. Simultaneously, the large amount of agglomeration of Ac-SEBS in PP seriously weakens the interfacial effect and may cause percolation, which causes the formation of a network of conductive pathways and a sharp increase of the conductive current. Moreover, because of the considerable accumulation of space charge, the inflection point of the transition from the ohmic current region to the SCLC region moves to the lower electric field.

## 4. Conclusions

In this paper, Ac-SEBS/PP blends with excellent electrical strength and mechanical toughness were prepared by the melt-blending process. From the scanning electron micrographs, it was found that Ac-SEBS with a 30% content could be well dispersed in the PP matrix when the acetylation degree was not high. According to the TGA experiment, Ac-SEBS/PP has good thermal stability. The DC breakdown field strength, conductive current density, space charge, and mechanical performance of Ac-SEBS/PP were mainly discussed in the paper. Acetophenone structural units on Ac-SEBS are able to absorb high-energy electrons through the keto-enol interconversion isomerization reaction and improve the DC breakdown strength of PP. The DC breakdown strength of Ac-SEBS/PP reached the highest level when the degree of acetylation was 4.6%, which is even a little higher than that of PP. Acetophenone structural units can reduce the conductive current density of the blends and inhibit space charge injection by capturing electrons. Besides, the highly elastic Ac-SEBS can enhance the tensile strength and elongation at break of PP to achieve the toughening effect. Incorporating Ac-SEBS with a suitable degree of acetylation into PP can maintain the excellent electrical properties and mechanical toughness of PP without damaging the raw material, providing an effective method for preparing environmentally friendly HVDC cable insulation materials. This research is expected to be applied to industrial production and enrich the material system of environmentally friendly HV cables.

## Figures and Tables

**Figure 1 materials-14-01596-f001:**
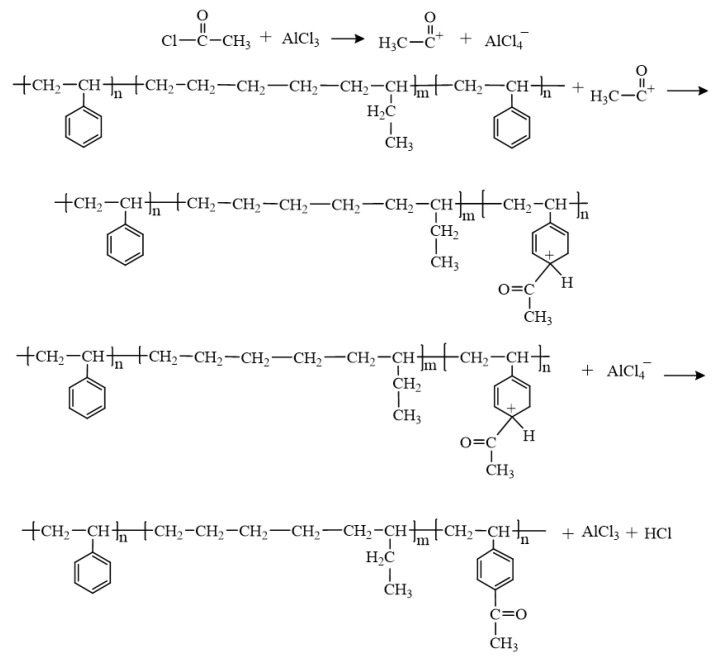
The preparation process of Ac-SEBS.

**Figure 2 materials-14-01596-f002:**
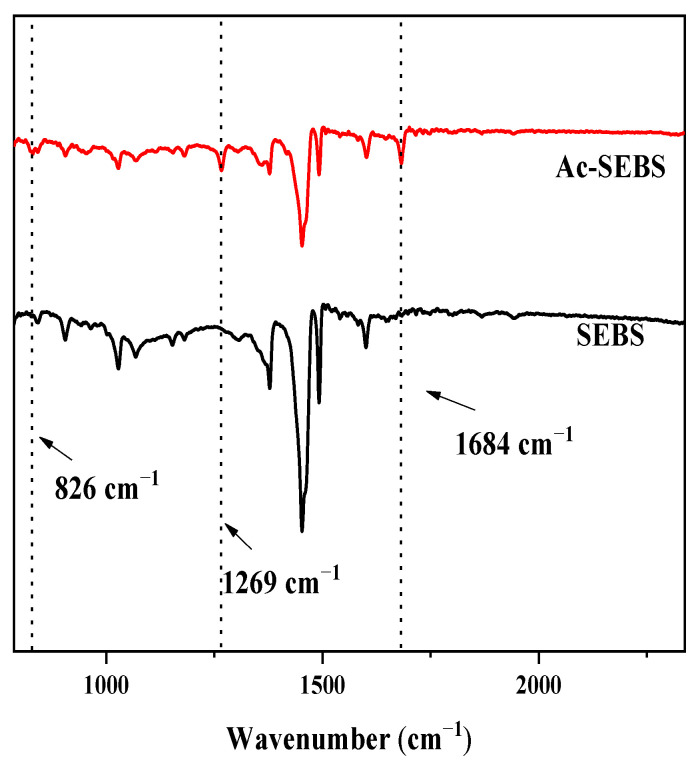
IR spectra of SEBS and Ac-SEBS.

**Figure 3 materials-14-01596-f003:**
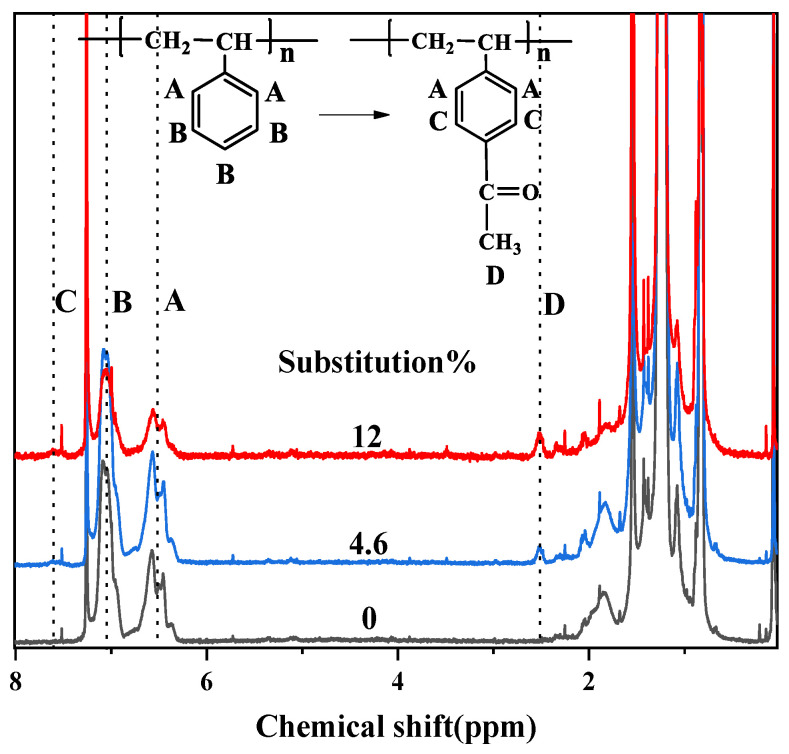
^1^H NMR spectrum of SEBS and Ac-SEBS.

**Figure 4 materials-14-01596-f004:**
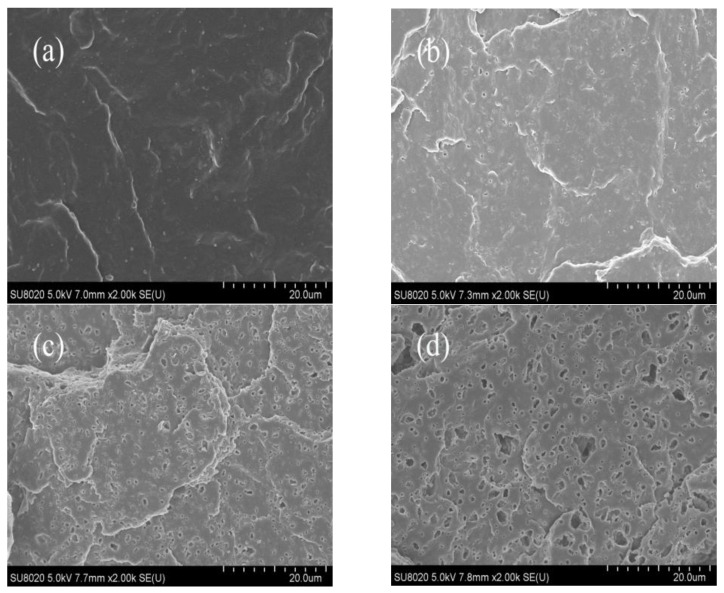
SEM images of Ac-SEBS/PP blends with different acetylation degrees of SEBS: (**a**): PP; (**b**): 0%; (**c**): 4.6%; (**d**): 12%.

**Figure 5 materials-14-01596-f005:**
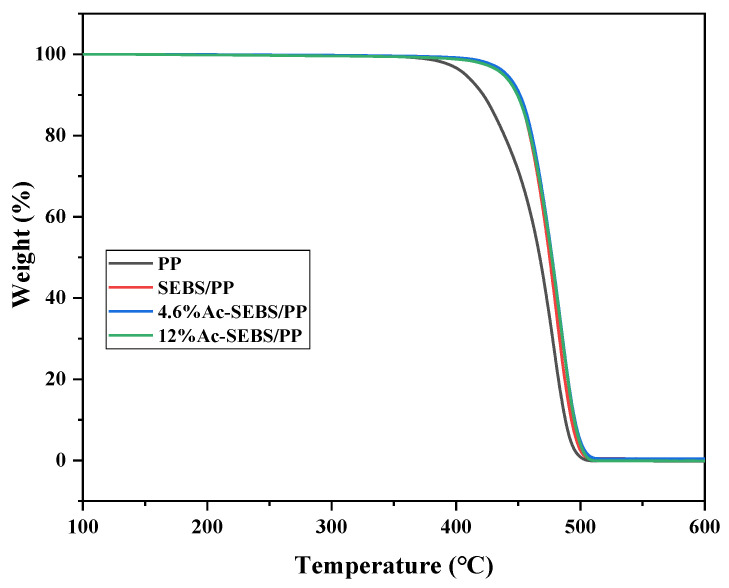
Thermogravimetric curves curve of PP and its blends.

**Figure 6 materials-14-01596-f006:**
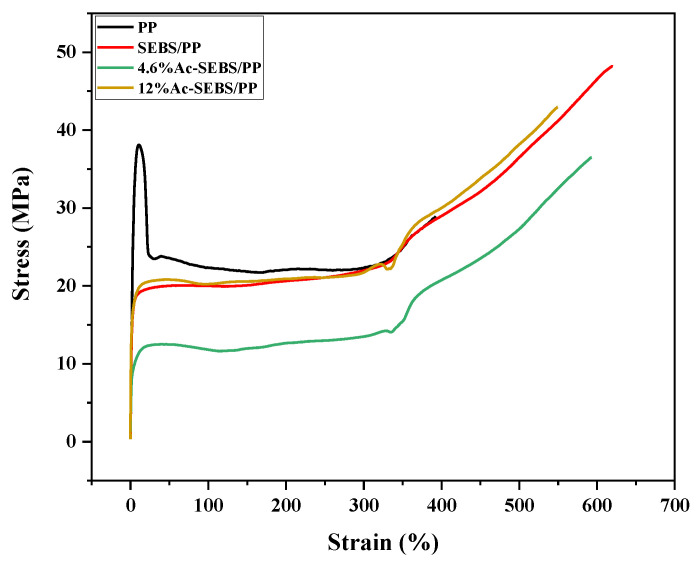
Stress–strain curve of PP and its blends.

**Figure 7 materials-14-01596-f007:**
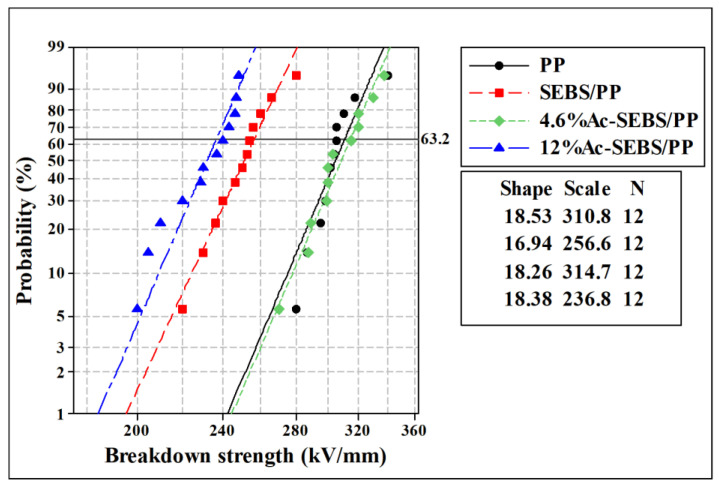
DC breakdown field strength of PP and its blends.

**Figure 8 materials-14-01596-f008:**
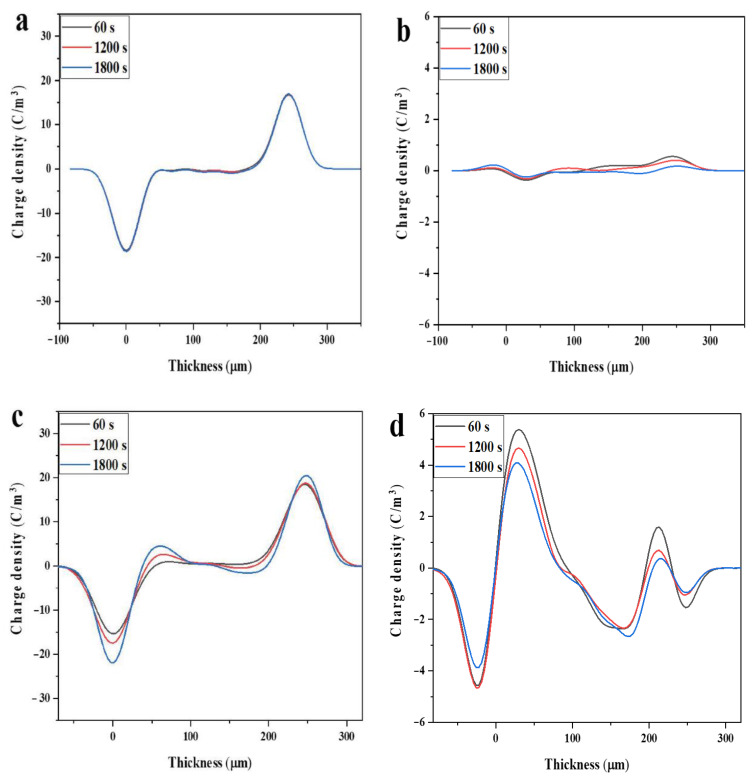
Space charge distribution (**a**) PP, (**c**) SEBS/PP, (**e**) 4.6% Ac-SEBS/PP, (**g**) 12% Ac-SEBS/PP and short circuit conditions (**b**) PP, (**d**) SEBS/PP, (**f**) 4.6% Ac-SEBS/PP, (**h**) 12% Ac-SEBS/PP.

**Figure 9 materials-14-01596-f009:**
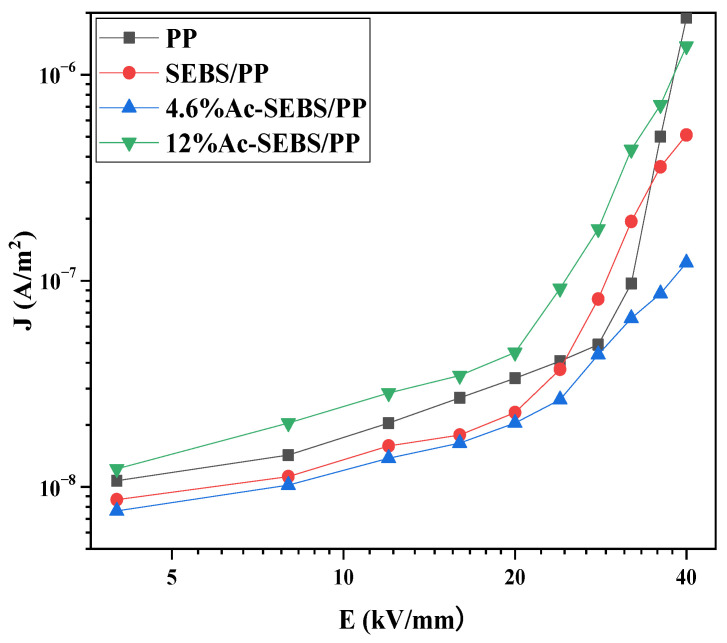
The conductive current of PP and its blends.

**Table 1 materials-14-01596-t001:** Summary of the tensile test data of the samples.

Sample	Elongation at Break (%)	Tensile Strength (MPa)
PP	392.6 ± 38.4	28.9 ± 2.2
SEBS/PP	619.8 ± 39.2	47.2 ± 1.8
4.6%Ac-SEBS/PP	593.0 ± 32.3	36.5 ± 1.7
12%Ac-SEBS/PP	549.3 ± 34.4	43.0 ± 2.0

## Data Availability

The data presented in this study are available on request from the corresponding author.

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
