# Peer review of "Effect of Acetylated SEBS/PP for Potential HVDC Cable Insulation"

_materials, 2021, doi:10.3390/ma14071596_

Round 1

Reviewer 1 Report

The authors open a very interesting research line based on the chemical modification of SEBS polymer matrix to produce a material useful for HVDC. This modified polymer (based on the acetylation of aromatic rings of PS) will be added to a matrix of PP. Al together will help to substitute PE derivatives as material to cover electrical wires.

The document is very well structured and the characterisation techniques are properly used. Anyhow, due the tensile stress resistance of the material must be optimal for such kind of applications I suggest to reflect in the document the standard specifications that the authors have used to test the material. If no standard has been used I recommend to repeat the experiments using the best adapted for the application.

Reviewer 2 Report

Authors presents method of acetylation of SEBS units and application of acetylated SEBS (Ac-SEBS) in blend with PP. Due to the excellent electrical strength such composition can be used in the high HVDC cable insulation. Results are interesting and well supported by experimental data. The materials and methods section are written well and provide detailed information.

The manuscript should be improved considering the following point:

Taking into consideration the application of Ac-SEBS/PP compositions in HVDC cable insulation the influence of the ageing on the electrical strength and mechanical properties should be analyzed. The results would be more interesting if authors can compare results before and after aging of the obtained materials. It is possible to predict changes in the material during aging/degradation time?

Yours sincerely

The Reviewer

Reviewer 3 Report

The paper presents an experimental work on the modification of PP via Ac-SEBS to improve the properties (mechanical and electrical) for a HVDC cable application. The paper is well presented and complete. I suggest publication following several minor corrections as:

L.61 : define « POE ».

L.61 : « et al. » missing.

L.62 : « et al. » missing.

L.89 : « et al. » missing.

L.136 : add vulcanization time.

L.144 : define « TMS ».

L.157 : how many repetitions for tensile tests ?

L.214 : Capital “S” for “Sub%”

L.223 : capital “T” for “The”.

L.245 : one decimal is enough…

Figure 5 : no need to put “%” on the values and axis title.

Figure 6 : Revise axes title “Stress (MPa)” and “Strain (%)”.

Table 1 : Too many digits: one decimal in enough for all values. Capital first letter for all parameters. The “tensile strength” reported looks more like the “tensile stress at break” !!!

L.289 : Fig. 7 (not 5).

L.365 : capital “F” for “Figure”.

L.381 : there is a word missing here, “structure” ?

Figure 9 : “J” must be defined somewhere. “/PP” is missing at the end of the last line in the legend.

The written English is good, but some parts (like the abstract) can be improved.

Always put a space between values and units everywhere.

I suggest to present (figure) the chemical structure of acetophenone.

The word “composites” is used throughout the document, but you actually mix polymers together… So it would be better to talk about “blends” or “compounds” here (there is no solid dispersed phase here.

Since it is reported that “It was analyzed that PP possesses excellent elastic modulus yet also has poor toughness” (L.264), I suggest to add data/discussion (Table 1) on tensile modulus and impact strength of all the samples produced.

I am curious how the Ac contents (0/4.6/12) were selected because there is significant differences between them…
